# Doxorubicin Impairs Smooth Muscle Cell Contraction: Novel Insights in Vascular Toxicity

**DOI:** 10.3390/ijms222312812

**Published:** 2021-11-26

**Authors:** Matthias Bosman, Dustin N. Krüger, Kasper Favere, Callan D. Wesley, Cédric H. G. Neutel, Birgit Van Asbroeck, Owen R. Diebels, Bart Faes, Timen J. Schenk, Wim Martinet, Guido R. Y. De Meyer, Emeline M. Van Craenenbroeck, Pieter-Jan D. F. Guns

**Affiliations:** 1Laboratory of Physiopharmacology, Faculty of Medicine and Health Sciences, Faculty of Pharmaceutical, Biomedical and Veterinary Sciences, Campus Drie Eiken, University of Antwerp, Universiteitsplein 1, B-2610 Antwerp, Belgium; dustin.kruger@uantwerpen.be (D.N.K.); kasper.favere@uantwerpen.be (K.F.); callan.wesley@uantwerpen.be (C.D.W.); cedric.neutel@uantwerpen.be (C.H.G.N.); owen.diebels@student.uantwerpen.be (O.R.D.); bart.faes@student.uantwerpen.be (B.F.); timen.schenk@student.uantwerpen.be (T.J.S.); wim.martinet@uantwerpen.be (W.M.); guido.demeyer@uantwerpen.be (G.R.Y.D.M.); pieter-jan.guns@uantwerpen.be (P.-J.D.F.G.); 2Research Group Cardiovascular Diseases, University of Antwerp, B-2610 Antwerp, Belgium; emeline.vancraenenbroeck@uantwerpen.be; 3Department of Internal Medicine, Faculty of Medicine and Health Sciences, Ghent University, C. Heymanslaan 10, B-9000 Ghent, Belgium; 4Department of Cardiology, Antwerp University Hospital (UZA), Drie Eikenstraat 655, B-2650 Edegem, Belgium; 5Research Group MOVANT, Department of Rehabilitation Sciences and Physiotherapy, Faculty of Medicine and Health Sciences, Campus Drie Eiken, University of Antwerp, Universiteitsplein 1, B-2610 Antwerp, Belgium; birgit.vanasbroeck@uantwerpen.be

**Keywords:** doxorubicin, cardio-oncology, arterial stiffness, endothelial dysfunction, vascular smooth muscle cell contraction, non-selective cation channel

## Abstract

Clinical and animal studies have demonstrated that chemotherapeutic doxorubicin (DOX) increases arterial stiffness, a predictor of cardiovascular risk. Despite consensus about DOX-impaired endothelium-dependent vasodilation as a contributing mechanism, some studies have reported conflicting results on vascular smooth muscle cell (VSMC) function after DOX treatment. The present study aimed to investigate the effects of DOX on VSMC function. To this end, mice received a single injection of 4 mg DOX/kg, or mouse aortic segments were treated ex vivo with 1 μM DOX, followed by vascular reactivity evaluation 16 h later. Phenylephrine (PE)-induced VSMC contraction was decreased after DOX treatment. DOX did not affect the transient PE contraction dependent on Ca^2+^ release from the sarcoplasmic reticulum (0 mM Ca^2+^), but it reduced the subsequent tonic phase characterised by Ca^2+^ influx. These findings were supported by similar angiotensin II and attenuated endothelin-1 contractions. The involvement of voltage-gated Ca^2+^ channels in DOX-decreased contraction was excluded by using levcromakalim and diltiazem in PE-induced contraction and corroborated by similar K^+^ and serotonin contractions. Despite the evaluation of multiple blockers of transient receptor potential channels, the exact mechanism for DOX-decreased VSMC contraction remains elusive. Surprisingly, DOX reduced ex vivo but not in vivo arterial stiffness, highlighting the importance of appropriate timing for evaluating arterial stiffness in DOX-treated patients.

## 1. Introduction

The anthracycline doxorubicin (DOX) is one of the most efficacious chemotherapeutic drugs for the treatment of a wide array of cancers [1]. However, DOX treatment has been associated with development of cardiotoxicity and eventually heart failure, which limits its clinical use [2,3,4,5,6]. While DOX-induced cardiotoxicity has been extensively investigated (reviewed elsewhere) [7], the toxic effects of DOX on the vasculature have been less considered. Epidemiological studies point towards accelerated vascular ageing in childhood cancer survivors, as evidenced by a higher incidence of atherosclerosis and hypertension [8,9,10]. Moreover, several studies have demonstrated that DOX increases arterial stiffness in cancer patients, both during and after treatment [11,12,13,14]. Arterial stiffness is associated with an increased risk of cardiovascular events and mortality, as it impacts arterial pressure, cardiac performance and perfusion [15,16,17].

Previously, it was shown that DOX impairs flow-mediated vasodilation in the brachial artery in patients shortly after DOX treatment, often referred to as endothelial dysfunction [18]. In addition, our research group demonstrated that 2 weeks of DOX administration to mice (4 mg/kg/week) acutely impaired endothelium-dependent vasodilation and consequently increased vascular tone, thereby actively augmenting arterial stiffness [19]. We, and others, identified endothelial cell loss and reduced endothelial nitric oxide synthase (eNOS) levels as potential mechanisms contributing to impaired endothelial function after DOX treatment [19,20]. Other studies have reported that DOX may cause uncoupling of eNOS, which is characterised by increased superoxide formation instead of nitric oxide production, thereby contributing to reactive oxygen species (ROS) generation [21,22].

Apart from the widely accepted toxic impact of DOX on endothelial cell (EC) function, DOX also affects vascular smooth muscle cell (VSMC) function [23,24,25]. There have been conflicting reports about the role of DOX in the alteration of VSMC reactivity. Gibson et al. reported decreased VSMC contraction in rat thoracic aorta rings 24 h after DOX treatment (15 mg/kg) [23]. Similarly, Olukman et al. demonstrated that DOX (20 mg/kg) attenuated VSMC contraction in rat aortic segments 1 week after administration [24]. Conversely, Shen et al. described increased VSMC contraction in mouse aortic rings after ex vivo DOX treatment (100 μM), which was attributable to increased Ca^2+^ release from the sarcoplasmic reticulum and Ca^2+^ entry in VSMCs [25]. In our previous study, we did not observe any direct effects of 2 weeks DOX treatment on VSMC contraction [19]. Hence, the exact effects of DOX on modulation of VSMC function remain unclear and may be dependent upon the dose or treatment protocol.

In this study, we aimed to investigate the effects of a single dose of DOX on VSMC function, including its contribution to arterial stiffness. A secondary goal was to evaluate whether the acute effects of DOX could be studied ex vivo in aortic rings. Such an ex vivo platform holds potential for mechanistic studies, as well as for future drug screening.

## 2. Results

### 2.1. A Single Dose of DOX Impairs VSMC Contraction and Endothelium-Dependent Vasodilation

There was a decrease in phenylephrine (PE)-induced contraction 16 h after in vivo DOX administration (Figure 1A). The PE-elicited contraction response, which comprises an initial transient and a subsequent tonic phase [26], was further investigated. DOX reduced the tonic but not the transient contraction in aortic segments after addition of 2 μM PE (Figure 1B).

Since the tonic contraction phase is dependent on Ca^2+^ influx [26], determining the increase in contraction force (Δ) at different time points offers an estimate of the effectiveness of Ca^2+^ influx in VSMCs. Here, we calculated the increase in force (Δ) by subtracting the force at 800 s from the force at 100 s during the PE-elicited contraction response. This increase in contraction force (Δ) was lower in the DOX-treated group compared with the vehicle-treated group (Figure 1C), which suggests that DOX impairs Ca^2+^ influx.

The contraction force remained lower in the DOX-treated group in the presence of Nω-nitro-L-arginine methyl ester (L-NAME) (Figure 1D). Of note, the PE contraction in panel D is the maximal amplitude of panel B, and these contraction amplitudes are higher than in panel A due to a decrease of basal nitric oxide (NO) over time [27]. Further, impaired acetylcholine (ACh)-induced endothelium-dependent relaxation was observed after DOX, while diethylamine NONOate (DEANO)-induced endothelium-independent relaxation was not affected (Figure 1E). These results suggest that DOX decreases VSMC contraction through a VSMC-specific mechanism.

### 2.2. Ex Vivo Incubation of Aortic Rings with DOX Leads to Reduced VSMC Contraction and Impaired Endothelium-Dependent Relaxation

To develop a drug testing platform for studying acute effects of DOX and to corroborate the in vivo DOX treatment findings, isolated aortic segments were isolated from mice and subsequently treated with DOX (1 μM) ex vivo for 16 h. Similar results were obtained under these experimental conditions. More specifically, PE-induced contraction was reduced after DOX treatment in a PE concentration response curve (Figure 2A) and contraction-over-time curve (Figure 2B). The increase in force (Δ) was lower for DOX-treated aortic segments compared with the vehicle group (Figure 2C). Under L-NAME conditions, the contraction remained decreased in the DOX group (Figure 2D), and there was impaired ACh-induced relaxation and unaltered DEANO-induced relaxation (Figure 2E).

### 2.3. DOX Does Not Modulate Ca^2+^ Release from the Sarcoplasmic Reticulum or Ca^2+^ Influx through Voltage-Gated Calcium Channels

To further investigate the molecular mechanisms of DOX-reduced VSMC contraction, the PE contraction (2 μM) was repeated under 0 mM Ca^2+^ Krebs Ringer and CaCl_2_ (3.5 mM Ca^2+^) conditions to specifically evaluate the transient contraction phase, caused by Ca^2+^ release from the sarcoplasmic reticulum, and the tonic contraction phase, characterised by VGCC- and NSCC-mediated Ca^2+^ entry, respectively. To avoid the possible influence of NO, experiments were performed under L-NAME conditions. Contraction mediated by Ca^2+^ release from the sarcoplasmic reticulum and decline in contraction due to Ca^2+^ uptake and Ca^2+^ efflux were not altered in response to DOX (Figure 3A). Upon addition of 3.5 mM Ca^2+^, thereby promoting Ca^2+^ entry through voltage-gated calcium channels (VGCCs) and non-selective cation channels (NSCCs), the resulting VSMC contraction was lower in the DOX-treated group compared with the vehicle group (Figure 3B). Subsequent addition of the VGCC blocker diltiazem (35 μM) revealed similar relaxation between the vehicle- and DOX-treated aortic segments (Figure 3C). VSMC contraction, elicited by different doses of K^+^ (10, 20, 30, 40 and 50 mM), was not altered after DOX treatment (Figure 3D). Moreover, incubation of aortic segments with the VGCC deactivator levcromakalim (1 μM), prior to addition of PE, resulted again in decreased tonic contraction in the DOX-treated group compared with the vehicle group (Figure 3E). These results suggest that DOX does not decrease VSMC contraction through molecular mechanisms responsible for intracellular Ca^2+^ release or Ca^2+^ uptake, Ca^2+^ efflux and VGCC-mediated Ca^2+^ entry.

Another possibility is that DOX affects NSCC-mediated contraction instead. Accordingly, further experiments were performed using several pharmacological agents blocking a variety of NSCCs, such as tranilast, SKF-96365, 2-aminoethoxydiphenyl borinate (2-APB) and 9-phenantrol. To avoid any possible inhibitory effect of NO on VGCCs and NSCCs, the following experiments were performed under L-NAME conditions.

Similar to the decrease in PE-elicited contraction (with L-NAME) under levcromakalim, DOX reduced the PE-induced VSMC contraction (2 μM) in aortic segments pre-incubated with diltiazem (35 μM) (Figure 4A). Subsequent addition of tranilast (100 μM), SKF-96365 (10 μM), 2-APB (20 μM) or 9-phenantrol (1 μM) resulted in a decline in VSMC contraction, which did not differ in absolute values (Figure 4B) but was higher in the DOX-treated group when expressed relative to its preceding contraction (Figure 4C).

Finally, to evaluate whether the decrease in VSMC contraction after DOX treatment was specific for the PE-elicited response, additional contraction agonists were used. DOX-treated aortic segments showed decreased VSMC contraction in response to 0.25 μM endothelin-1 (Figure 5A) but not after 2 μM angiotensin II (Figure 5B), 2 μM serotonin (Figure 5C) and 50 mM K^+^ (Figure 5D) stimulation.

### 2.4. A Single Dose of DOX Decreases Ex Vivo Arterial Stiffness but Not In Vivo Arterial Stiffness and Blood Pressure

The possible translation of the aforementioned findings to arterial stiffness was investigated by evaluation of ex vivo and in vivo arterial stiffness.

The ROTSAC set-up was used for ex vivo evaluation of arterial stiffness, 16 h after in vivo DOX administration (4 mg/kg). Under Krebs Ringer conditions, the Peterson’s modulus (Ep), a measure for arterial stiffness, was lower in the DOX group, especially at higher pressures (Figure 6A). This effect was less pronounced in the presence of PE combined with L-NAME (Figure 6B). In the presence of DEANO, which removes VSMC tonus, DOX decreased Ep (Figure 6C) in a pressure-dependent way.

In vivo arterial stiffness was investigated by measuring the abdominal aorta pulse wave velocity (aaPWV) with ultrasound imaging as well as carotid–femoral pulse wave velocity (cfPWV) with tonometry, 16 h after injecting a single dose of DOX. Blood pressure and left ventricular ejection fraction (LVEF) were measured as well. DOX did not change aaPWV (Figure 6D), cfPWV (Figure 6E), blood pressure (Figure 6F,G) and LVEF (Figure 6H).

## 3. Discussion

It has been recently recognised that DOX impairs vascular function [11,12,13,14,19], yet the exact mechanisms are not fully elucidated yet. While there is consensus about the detrimental effects of DOX on endothelium-dependent vasodilation [19,20,21,22], several studies have reported conflicting results on VSMC contraction [23,24,25]. The present study therefore aimed to investigate VSMC function after a single DOX dose in mice and how it contributes to arterial stiffness.

DOX decreased PE-induced VSMC contraction and impaired ACh-induced endothelium-dependent relaxation in both the in vivo and ex vivo treatment setting. Impaired ACh-induced relaxation was in line with previous reports [19,20,21]. Endothelial cell loss and reduced eNOS-expression contribute to DOX-induced endothelial dysfunction [19,20,21], which leads to decreased NO bioavailability [28]. This, in turn, results in increased vascular tone and augmented arterial stiffness [29,30]. Contradictory, in the present study, we observed a reduction in VSMC contraction following DOX treatment, even after L-NAME addition. L-NAME inhibits eNOS function, thereby excluding the influence of NO in modulating vascular tone. Accordingly, our data suggest that DOX impairs VSMC contraction through a VSMC-specific mechanism.

For the convenience of the reader, Figure 7 illustrates the different mechanisms involved in PE-elicited contraction. Moreover, the different experimental conditions and compounds that were used to elucidate the contributing mechanisms to DOX-induced decrease in VSMC contraction are schematically illustrated in Figure 7 [26].

VSMC sensitivity for NO was not affected by DOX, as evidenced by similar endothelium-independent relaxation with DEANO. It is thus most likely that the cyclic guanosine monophosphate (cGMP) pathway is not targeted by DOX.

Further experiments demonstrated that DOX did not alter the initial inositol trisphosphate (IP_3_)-mediated contraction under 0 mM Ca^2+^ conditions, suggesting that DOX does not affect Ca^2+^ release from the sarcoplasmic reticulum. Moreover, the subsequent decline in force was similar in the vehicle- and DOX-treated groups. This indicates that DOX does not affect Ca^2+^ uptake and Ca^2^ efflux, mediated, for example, by the sarco-endoplasmic reticulum and plasmalemma Ca^2+^–ATPase pumps, respectively. However, we observed that DOX particularly reduced the tonic contraction phase, which may point to altered Ca^2+^ influx, either through VGCCs or NSCCs. Relative relaxation of PE-contracted aortic segments with diltiazem, a VGCC-blocker, was not altered after DOX treatment. Under 10–50 mM K^+^ conditions, depolarisation of the membrane potential will promote VGCC-mediated contraction. VSMC contraction with different doses of K^+^ (10, 20, 30, 40 and 50 mM) was similar between the vehicle- and DOX-treated group. Moreover, after pre-incubation of aortic segments with levcromakalim and diltiazem, tonic VSMC contraction was lower in the DOX-treated group compared with the vehicle-treated group. These findings indicate that VGCCs are not the molecular mechanism explaining the reduced contractions in the presence of DOX. The attenuated VSMC contraction is therefore possibly attributable to DOX-induced alteration of NSCC function.

Recently, it was found that DOX can inhibit the transient receptor potential vanilloid 2 (TRPV2) channel, a member of the extensive family of NSCCs, in cancer cells by permeating into the channel pore, thereby blocking it [32,33]. Transient receptor potential (TRP) channels are involved in tumorigenesis, tumour proliferation and tumour migration [32,34,35,36,37]. Since the discovery that Ca^2+^ influx plays a crucial role in cancer development [38], targeting corresponding mediators, such as TRP channels, has proven to be an effective strategy in cancer treatment [35,37,39]. Importantly, TRP channels play a pivotal role in VSMC contraction, where they participate in Ca^2+^ influx and Ca^2+^ homeostasis [40]. Therefore, it is conceivable that DOX, as a chemotherapeutic drug, also inhibits TRP channels in aortic VSMCs in a manner that is similar to tumour cells, thereby contributing to reduced VSMC contraction and vascular toxicity.

To investigate the possible role of DOX in targeting members of the NSCC family, several NSCC blockers, such as tranilast, SKF-96365, 2-APB and 9-phenantrol were evaluated. We limited our selection of NSCC blockers to inhibitors of TRP channels, which have been proven to be involved in VSMC contraction [40]. One should be cautious in interpreting the data from the experiments evaluating the role of DOX in NSCC function because high molecular structure homology between the different subtypes of TRP channels limits the specificity of corresponding NSCC blockers [40]. Accordingly, we selected two specific TRPV2 inhibitors—namely, tranilast and SKF-96365 (as stated by literature) [41,42]—and the rather general TRP inhibitor 2-APB. Finally, the TRPM4 blocker 9-phenantrol was also used due to a recently identified role of TRPM4 in breast cancer [43].

Surprisingly, DOX treatment resulted in a greater decline in VSMC contraction in the presence of tranilast, SKF-96365, 2-APB and 9-phenantrol, when expressed relative to the preceding contraction magnitude, showing the contribution of TRP channels to PE contractions. This paradoxical finding suggests that TRP channels play a relatively larger role in mediating contraction in DOX-treated aortic segments, and therefore may not be responsible for the decreased VSMC contraction after DOX. Instead, DOX may target other NSCCs that result in attenuated contraction. However, the lack of knowledge regarding the exact role of TRP channels in vascular function, the high diversity of the NSCC family and the interaction between NSCCs mutually [40] constitutes an important limitation in the current study. Further research to elucidate the mechanistic aspects of NSCCs in vascular biology and also how DOX targets these pathways would be interesting, yet such research lies beyond the scope of the present work.

Other agonists, such as endothelin-1, angiotensin II and serotonin were subsequently used to determine whether our observations were specific for PE-involved pathways. Endothelin-1, angiotensin II and serotonin were selected as representative agonists for NSCC-, sarcoplasmic reticulum- and VGCC-mediated contraction, respectively. More specifically, endothelin-1 mainly induces vasoconstriction through VGCC- and NSCC-mediated Ca^2+^ influx, while angiotensin II mediates a phasic contraction through Ca^2+^ release from intracellular Ca^2+^ stores, such as the sarcoplasmic reticulum [44]. Serotonin promotes VSMC constriction through Ca^2+^ release from the sarcoplasmic reticulum in a first transient phase and through primarily VGCC-mediated Ca^2+^ influx in a second tonic phase [45]. DOX decreased VSMC contraction in response to endothelin-1, but not in the presence of angiotensin II or serotonin. These observations further corroborate that Ca^2+^ release from intracellular Ca^2+^ stores as well as Ca^2+^ influx through VGCCs are not targeted by DOX but that DOX may decrease VSMC contraction through interference with a specific, yet currently unknown, NSCC. Hence, DOX reduces VSMC contraction in a PE receptor-independent way.

Surprisingly, a single dose of DOX decreased ex vivo arterial stiffness at higher pressures, which is in contrast with our previous report where two weeks DOX treatment increased VSMC tone and arterial stiffness [19]. It is important to note that endothelial and VSMC function together actively regulate vascular tone, thereby modulating arterial stiffness [29,30]. As such, the occurrence of both impaired endothelium-dependent relaxation and diminished VSMC contraction may have a counteracting effect on arterial stiffness. Another possible explanation for the discrepancies regarding VSMC contraction in the previous [19] and current study lies in the timing for evaluating arterial stiffness. It is conceivable that, in the first acute phase, DOX reduces both VSMC contraction and contributes to endothelial dysfunction, leading to a decrease in arterial stiffness. In a later phase, DOX-induced VSMC impairment reverses, most likely due to clearance of DOX from the vascular system over time, while endothelial dysfunction persists. This, in turn, will increase VSMC tone, thereby actively augmenting arterial stiffness.

In contrast with our previously reported work that DOX increased arterial stiffness after two weeks [19], we found no change in in vivo arterial stiffness and blood pressure 16 h after DOX administration, possibly due to opposing effects of DOX on endothelial and VSMC function. Based on our previous and current data, these findings may have clinical consequences. More specifically, the unchanged and increased arterial stiffness after 16 h and two weeks of DOX treatment, respectively, may highlight the importance of appropriate timing for evaluating arterial stiffness in DOX-treated patients.

Previous studies have reported decreased VSMC contraction after DOX treatment [23,24]. In these reports, it is hypothesised that DOX-reduced contraction was mediated by diminished α1-adrenergic receptor expression or by ROS-induced VSMC death. While others reported similar results [9,46,47,48,49], the data in the current study do not support these proposed mechanisms. In the case of reduced α1-adrenergic receptor expression, the entirety of the PE-contraction response would be affected, including the IP_3_-mediated contraction, which was not observed. Given the fact that DOX also reduced the endothelin-1-elicited VSMC contraction, it further supports that DOX-decreased contraction occurs independent of α1-adrenergic receptors. Furthermore, depolarisation-induced contraction would be mitigated in the event of VSMC death [50,51], yet contraction under 50 mM K^+^ conditions was not affected, which excludes the occurrence of VSMC death. Hence, our data indicate that DOX decreases VSMC contraction without affecting α1-adrenergic receptor expression or contributing to VSMC death.

The difference in the used dosage of DOX in the present study and the aforementioned literature [23,24,25] may provide an explanation for the divergent mechanisms. The recommended maximal cumulative DOX dose in patients is 450 mg/m^2^, which corresponds to 12 mg/kg (assuming a body surface area of 1.9 m^2^ and a body weight of 70–75 kg) [52,53]. Additionally, the maximum tolerated dose (MTD) and median lethal dose (LD_50_) of DOX in *C57BL6/J* mice are both 10 mg/kg [54]. Previous reports on reduced VSMC contraction following DOX treatment used a single high dose (>10 mg/kg), which could induce VSMC death [21,23,24]. In the current work, however, a lower DOX dose (4 mg/kg) was used for optimal comparison with previous work, which may affect VSMC function differently.

As a secondary goal, we aimed to develop an ex vivo experimental platform for mechanistic studies and for drug screening. In this experimental setting, a DOX dose of 1 μM was used, which is within therapeutic range, since plasma concentrations of DOX have been reported to fluctuate between 0.3 and 1 μM in patients [48,55]. Since the observed changes in vascular reactivity in response to DOX were similar in vivo and ex vivo, the platform holds potential for future drug screening.

In conclusion, the present study reports decreased VSMC contraction with coinciding endothelial dysfunction after DOX treatment (4 mg/kg). The decrease in contraction does not relate to alterations in intracellular Ca^2+^ release or Ca^2+^ influx through VGCCs, but points towards a modulating role for DOX in NSCC function. Experiments with TRP blocking agents failed to pinpoint the exact mechanism causing reduced VSMC contractions, suggesting that DOX may target other NSCCs instead. Importantly, the observation of unchanged in vivo arterial stiffness or blood pressure, possibly due to counteracting effects of DOX on EC and VSMC function, may highlight the importance of timing for evaluation of DOX-induced arterial stiffness in patients.

## 4. Materials and Methods

### 4.1. Animals and Ethical Approval

Male *C57BL/6J* mice (Charles River, Ecully, France) with an age of 10 to 12 weeks and a body weight between 26 and 30 g were used for all experiments. Mice were housed in the animal facility of the University of Antwerp in standard cages with 12–12 h light–dark cycles with access to regular chow and water ad libitum. To avoid the influence of female hormone confounding factors, male mice were chosen. All experiments in this work were approved by the Ethical Committee of the University of Antwerp (dossier 2019-34) and conformed with the ARRIVE guidelines, with the *Guide for the Care and Use of Laboratory Animals* published by the US National Institutes of Health (NIH Publication no. 85–23, revised 1996) and with the Belgian Royal Decree of 2013.

### 4.2. DOX Treatment and Experimental Workflow

The current study composed two distinct experimental parts—namely, a part where DOX was administered to mice (in vivo DOX treatment), and a second part where aortic segments were first isolated from mice and subsequently either treated with vehicle or DOX (ex vivo DOX treatment).

For in vivo DOX treatment: *C57BL/6J* mice received a single intraperitoneal injection of 4 mg/kg DOX (Adriamycin^®^ (2 mg/mL), Pfizer, Puurs, Belgium) or 10 mL/kg of a 0.9% NaCl solution (B. Braun, Machelen, Belgium) as vehicle. DOX was diluted in a 0.9% NaCl solution on the day of injection. Based on other studies that investigated acute DOX effects [21,23,24,56], mice were sacrificed 16 h later for vascular reactivity evaluation in organ baths with isometric transducer. To this end, mice were intraperitoneally injected with sodium pentobarbital (100 mg/kg; Sanofi, Machelen, Belgium), followed by perforation of the diaphragm (when under deep anaesthesia). The thoracic aorta was carefully dissected and cut into six segments of 2 mm length (i.e., TA0 to TA5), with the crossing of the diaphragm as the reference point for the sixth segment (TA5). Next, TA3, TA4 and TA5 segments were mounted between two hooks of an organ bath set-up (10 mL) filled with Krebs Ringer solution (37 °C, 95% O2/5% CO2, pH 7.4) for vascular reactivity and arterial stiffness evaluation. The Krebs Ringer solution contained (in mmol/L) NaCl 118, KCl 4.7, CaCl_2_ 2.5, KH_2_PO_4_ 1.2, MgSO_4_ 1.2, NaHCO_3_ 25, CaEDTA 0.025 and glucose 11.1.

For ex vivo DOX treatment: Healthy wild-type *C57BL/6J* mice were first sacrificed and the aorta was subsequently isolated, as described above. Following dissection, aortic segments were immediately transferred to Dulbecco’s modified Eagle medium (DMEM) medium (Thermo Fisher Scientific, Geel, Belgium) supplemented with 1% penicillin/streptomycin (Thermo Fisher Scientific, Geel, Belgium). Aortic segments were randomly divided and received either phosphate-buffered saline solution (PBS) as vehicle (1:100) or DOX (1 μM) for 16 h. DOX was diluted in PBS on the day of the experiment. After 16 h, aortic segments were mounted between two hooks of an organ bath set-up (10 mL) filled with Krebs Ringer solution (37 °C, 95% O_2_/5% CO_2_, pH 7.4) for vascular reactivity evaluation.

### 4.3. High-Frequency Ultrasound Imaging

Ultrasound imaging was performed in anaesthetised mice under 1.5–2.5% (*v*/*v*) isoflurane (Forene; Abbvie, Wavre, Belgium) using a high-frequency ultrasound system (Vevo2100, VisualSonics, Toronto, Canada). Images were only acquired when heart rate and body temperature met the inclusion criteria, i.e., 550 ± 50 beats/min and 37 ± 1 °C, respectively. M-mode images were obtained for determination of cardiac parameters using a 24 MHz transducer. LVEF was subsequently calculated using measurements of three consecutive M-mode cycles with Vevo LAB Software (Version 3.2.0, VisualSonics, Toronto, Canada). In the same session, abdominal aorta PWV (aaPWV) was determined according to the method developed by Di Lascio et al. with a 24 MHz transducer [57]. Briefly, pulse wave Doppler tracing was used to measure aortic flow velocity (V). Immediately thereafter, aortic diameter (D) was measured on 700 frames-per-second B-mode images of the abdominal aorta in EKV imaging mode. The ln(D)-V loop method was then applied to calculate aaPWV, using MATLAB v2014 software (MathWorks, Eindhoven, the Netherlands).

### 4.4. Applanation Tonometry

cfPWV was determined in anaesthetised mice under 1.5–2.5% (*v*/*v*) isoflurane (Forene; Abbvie, Wavre, Belgium), as previously described by our research group [58]. In brief, two pulse tonometers (SPT-301, Millar Instruments, Wokingham, United Kingdom) were applied on the skin using a micromanipulator. Carotid–femoral transit time (Δt) was determined using the time difference between the foot of carotid and femoral artery pulses (foot-to-foot method). The foot of the pressure wave was defined as the second derivative maximum. Twenty consecutive pulses with sufficient amplitude and a reproducible waveform were analysed; pulses that interfered with respiratory movement peaks were excluded.

### 4.5. Blood Pressure Evaluation

Systolic and diastolic blood pressure were determined non-invasively in restrained, awake mice using a tail-cuff system with programmed electrosphygmomanometer (Coda, Kent Scientific Corporation, Torrington, United States of America). To reduce stress and variability during the procedure, animals were trained for two days prior to the actual measurements.

### 4.6. Evaluation of Vascular Reactivity

For clarity, the standard experimental protocol for the in vivo and ex vivo DOX treatment experiments is illustrated in Appendix A, respectively. Aortic segments were mounted at a preload of 20 mN. Since we have previously shown that basal NO declines over time [27], the experimental protocol for in vivo DOX-treated aortic segments was started exactly 70 min after puncture of the diaphragm to minimise time-dependent biases. Ex vivo DOX-treated aortic segments were also mounted at a preload of 20 mN, and the experimental protocol was started 20 min thereafter to allow optimal stabilisation. VSMC contraction was evaluated by adding cumulative concentrations of PE (3 nM–3 μM), an α1-adrenergic receptor agonist. Next, endothelium-dependent relaxation was investigated by addition of cumulative concentrations of ACh (3 nM–10 μM), a muscarinic receptor agonist.

The PE-elicited contraction response was further investigated by incubating the aortic segments in the organ bath with a single PE-dose of 2 μM for 15 min. Once the contraction was stable after 15 min, L-NAME (300 μM) was subsequently added, further increasing contraction (PE + L-NAME contraction). L-NAME is an eNOS blocker, which inhibits NO production and allows evaluation of the involvement of the endothelial cell layer in regulating VSMC contraction and tone. After 20 min, cumulative concentrations of the exogenous NO-donor DEANO (0.3 nM–10 μM) were added to the organ bath to evaluate endothelium-independent relaxation of VSMCs through the cGMP-mediated pathway.

The contraction via IP_3_-mediated Ca^2+^ release from the sarcoplasmic reticulum was investigated by incubating aortic segments with Ca^2+^-free Krebs Ringer solution (0 mM Ca^2+^) for three minutes, followed by addition of PE (2 μM) for three minutes. The 0 mM Ca^2+^ Krebs Ringer solution contained (in mmol/L): NaCl 118, KCl 4.7, CaCl_2_ 0, KH_2_PO_4_ 1.2, MgSO_4_ 1.2, NaHCO_3_ 25, CaEDTA 0.025, EGTA 1.0 and glucose 11.1. An extracellular 0 mM Ca^2+^ environment will prevent Ca^2+^ influx during PE contraction, which will shift contraction to solely Ca^2+^ release from the sarcoplasmic reticulum [26]. Next, CaCl_2_ (3.5 mM) was added to restore Ca^2+^-containing Krebs, which resulted in contraction in a similar way as the PE-elicited contraction response [26]. Finally, the involvement of VGCCs was determined by adding a single, high concentration (35 μM) of diltiazem, a voltage-gated Ca^2+^-channel blocker. This resulted in relaxation, which offered an estimate of the amount of contraction that was attributable to VGCCs.

To determine the role of VSMC membrane potential in modulating contraction, a dose–response of K^+^ (10 mM, 20 mM, 30 mM, 40 mM and 50 mM) was performed in the organ bath. A high dose of K^+^ causes depolarisation of the membrane potential, thereby activating VGCC-mediated Ca^2+^ influx, which eventually results in contraction [26,59]. Finally, the VGCC blockers levcromakalim (1 μM) and diltiazem (35 μM) and the NSCC blockers tranilast (100 μM), SKF-96365 (10 μM), 2-APB (20 μM) and 9-phenantrol (1 μM) were used to delineate the involvement of NSCCs in the PE-elicited contraction response. Levcromakalim hyperpolarises the VSMC membrane potential, thereby deactivating VGCCs, which will consequently inhibit Ca^2+^ influx via these channels [26,60]. These experiments were performed in the presence of L-NAME to exclude the influence of NO on VGCCs and NSCCs. Prior to the addition of a NSCC blocker, PE-contraction was elicited in the presence of diltiazem, thereby promoting Ca^2+^ influx through NSCCs. The protocol for investigation of NSCCs in modulating contraction is illustrated in Appendix A.

### 4.7. Evaluation of Arterial Stiffness in ROTSAC

Ex vivo assessment of arterial stiffness was performed using the ROTSAC, as previously described [19,61]. In brief, segments (TA3) were continuously stretched between alternating preloads corresponding to “systolic” and “diastolic” transmural pressures. Calibration of the set-up allows calculation in real-time of the “systolic” and “diastolic” pressure and the Peterson’s modulus (Ep), a measure of arterial stiffness, based on LaPlace’s equation. Ep was calculated as follows: Ep = D_0_ * ΔP/ΔD, with ΔP = difference in pressure (kept constant at 40 mmHg), D_0_ = “diastolic” diameter and ΔD = the change in diameter between “diastolic” and “systolic” pressure. The ROTSAC protocol included the evaluation of arterial stiffness (Ep) at different pressures (i.e., 60–100 until 220–260 mmHg with 20 mmHg intervals). The contribution of VSMC tonus was investigated by adding a high concentration (2 μM) of PE combined with L-NAME (300 μM).

### 4.8. Chemical Compounds

DOX (Adriamycin^®^, 2 mg/mL) was purchased from Pfizer (Puurs, Belgium). PE, L-NAME, ACh, DEANO, diltiazem, tranilast, SKF-96365, 9-phenantrol, serotonin hydrochloride and angiotensin II (human) were obtained from Sigma-Aldrich (Overijse, Belgium). Levcromakalim and 2-APB were purchased from TOCRIS (Bristol, United Kingdom). Finally, endothelin-1 was supplied by Alexis^®^ Biochemicals (Enzo Life Sciences, Zandhoven, Belgium).

### 4.9. Statistical Analysis

All results are expressed as the mean ± standard error of the mean (SEM). Statistical analyses were performed using GraphPad Software (Prism 9—Version 9.2.0; Graphpad, California, United States of America). A *p*-value < 0.05 was considered to be statistically significant.

## Figures and Tables

**Figure 1 ijms-22-12812-f001:**
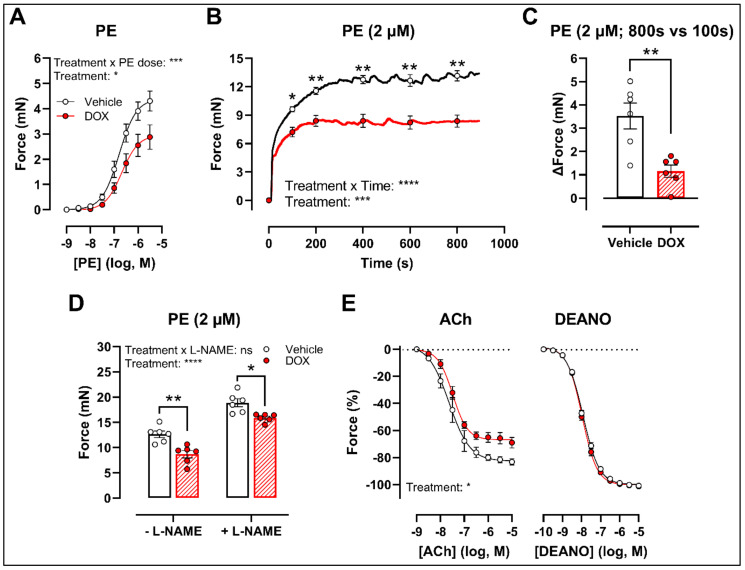
VSMC contraction and relaxation in aortic segments, 16 h after in vivo DOX administration. PE-induced contraction was decreased in aortic segments after in vivo DOX treatment (**A**). The PE-elicited contraction response was reduced after DOX treatment, which suggests perturbed tonic contraction (**B**). The increase in force between 800 s and 100 s was lower in the DOX group, indicating that DOX impairs Ca^2+^ influx (**C**). The DOX-induced reduction in contraction persisted under L-NAME conditions (**D**). DOX impaired ACh-induced relaxation, but relaxation of aortic segments in response to DEANO did not change (**E**). For (**A**,**B**,**E**), repeated measures two-way ANOVA with Šidàk’s multiple comparisons test. For (**C**), unpaired *t*-test. For (**D**), two-way ANOVA with Tukey’s multiple comparisons test. *, **, ***, **** *p* < 0.05, 0.01, 0.001, 0.0001; *n* = 6 for each group.

**Figure 2 ijms-22-12812-f002:**
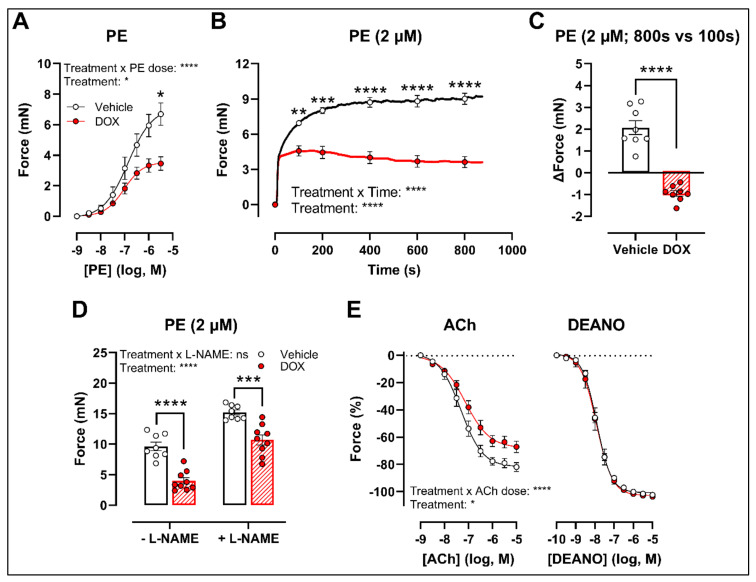
VSMC contraction and relaxation in aortic segments after 16 h of ex vivo DOX treatment. Following 16 h of ex vivo DOX treatment, VSMC contraction was decreased in response to PE (**A**,**B**). In the DOX-treated aortic segments, there was a lower increase in force between 800 s and 100 s (**C**). The DOX-induced decrease in contraction was still observed in the presence of L-NAME (**D**). ACh-induced relaxation, but not DEANO-induced relaxation, was impaired in response to DOX (**E**). For (**A**,**B**,**E**), repeated measures two-way ANOVA with Šidàk’s multiple comparisons test. For (**C**), unpaired *t*-test. For (**D**), two-way ANOVA with Tukey’s multiple comparisons test. *, **, ***, **** *p* < 0.05, 0.01, 0.001, 0.0001; *n* = 8 for vehicle group and *n* = 9 for DOX group.

**Figure 3 ijms-22-12812-f003:**
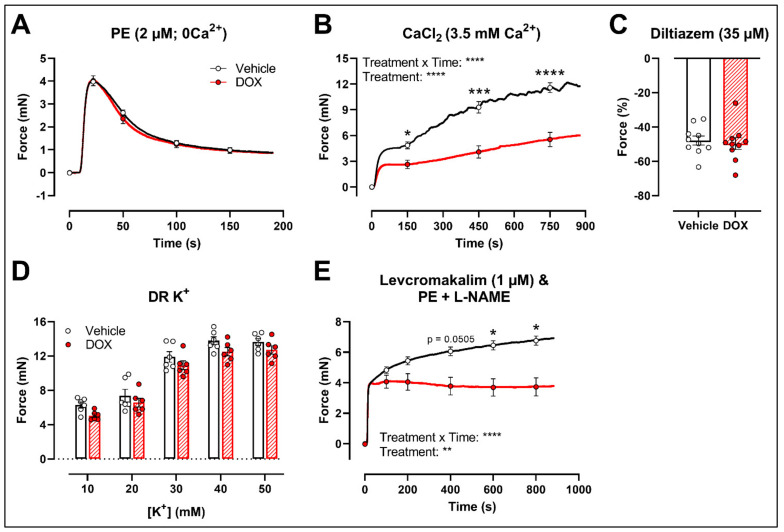
Contraction of aortic segments mediated by Ca^2+^ release from the sarcoplasmic reticulum and Ca^2+^ entry through VGCCs and NSCCs after 16 h of ex vivo DOX treatment. (**A**) Contraction through Ca^2+^ release from the sarcoplasmic reticulum in response to PE (2 μM) in the absence of extracellular Ca^2+^ (0 mM Ca^2+^). (**B**) Addition of 3.5 mM Ca^2+^ after 0 mM Ca^2+^ PE-elicited contraction. (**C**) Addition of 35 μM diltiazem following stable CaCl_2_-mediated contraction. (**D**) Dose–response of K^+^-elicited contraction (10, 20, 30, 40 and 50 mM). (**E**) PE-induced contraction (with L-NAME) under levcromakalim (1 μM). DOX did not change contraction through Ca^2+^ release from the sarcoplasmic reticulum, nor Ca^2+^ uptake and Ca^2+^ efflux after 16 h of ex vivo DOX treatment (**A**), but VSMC contraction was decreased in the DOX-treated group after CaCl_2_ addition (**B**). There was no difference in VSMC relaxation between the groups after blocking of VGCCs with diltiazem (**C**), nor in the contraction magnitude in response to different K^+^ doses (**D**). DOX decreased PE-induced contraction (with L-NAME) in the presence of levcromakalim (**E**). For (**A**,**B**,**D**,**E**), repeated measures two-way ANOVA with Šidàk’s multiple comparisons test. For (**C**), unpaired *t*-test. *, ***, **** *p* < 0.05, 0.001, 0.0001; *p* > 0.05 in (**A**,**C**,**D**); (**A**–**C**) *n* = 10 in each group; (**D**,**E**) *n* = 6 in each group.

**Figure 4 ijms-22-12812-f004:**
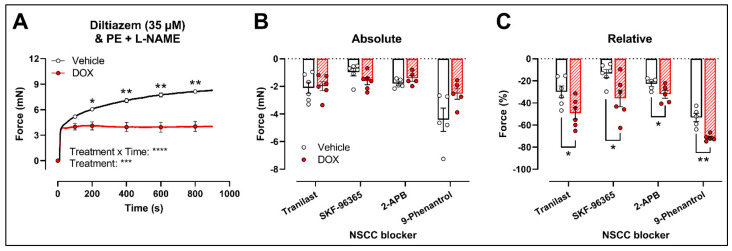
Evaluation of the involvement of members of the NSCC family in decreased PE-induced contraction in aortic segments after 16 h of ex vivo DOX treatment. (**A**) PE-induced contraction (2 μM, with L-NAME) under diltiazem (35 μM). (**B**,**C**) Absolute and relative decline in PE-induced contraction (with L-NAME and diltiazem) under different NSCC-blocker conditions. DOX decreased PE-induced contraction (with L-NAME) in the presence of diltiazem (**A**). In addition, the DOX-treated group did not display a significant change in VSMC contraction in absolute values (**B**) but showed a greater decline in VSMC contraction when expressed relative to its preceding contraction magnitude under tranilast (100 μM), SKF-96365 (10 μM), 2-APB (20 μM) and 9-phenantrol (1 μM) (C). For (A), repeated measures two-way ANOVA with Šidàk’s multiple comparisons test. For (**B**,**C**), unpaired *t*-test for each NSCC blocking condition. *, **, ***, **** *p* < 0.05, 0.01, 0.001, 0.0001; *p* > 0.05 in (**B**); *n* = 6 in each group.

**Figure 5 ijms-22-12812-f005:**
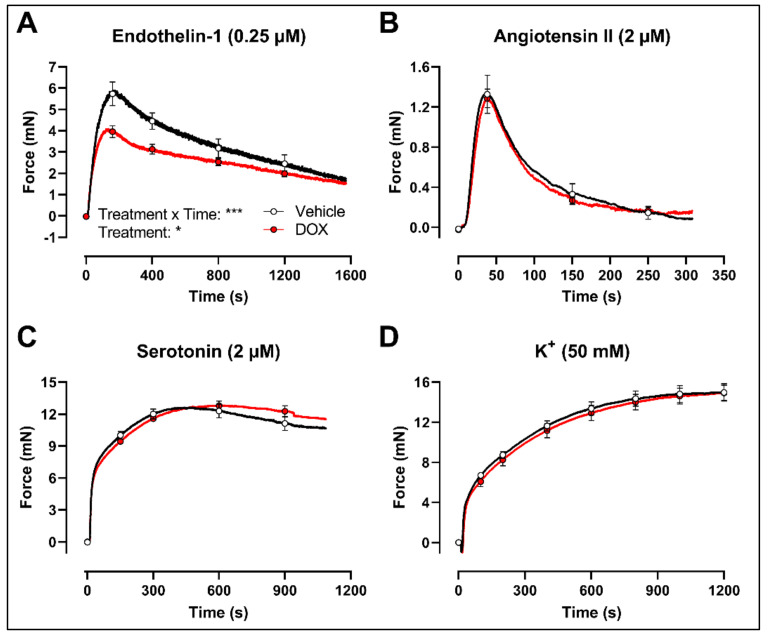
Contraction of aortic segments with endothelin-1, angiotensin II, serotonin and K^+^ after 16 h of ex vivo DOX treatment. DOX decreased VSMC contraction in response to 0.25 μM endothelin-1 (**A**) but not after stimulation with 2 μM angiotensin II (**B**), 2 μM serotonin (**C**) and 50 mM K^+^ (**D**). For (**A**–**D**), repeated measures two-way ANOVA with Šidàk’s multiple comparisons test. *, *** *p* < 0.05, 0.001; *p* > 0.05 in (**B**–**D**); For (**A**–**C**), *n* = 6 in each group; for (**D**), *n* = 10 in each group.

**Figure 6 ijms-22-12812-f006:**
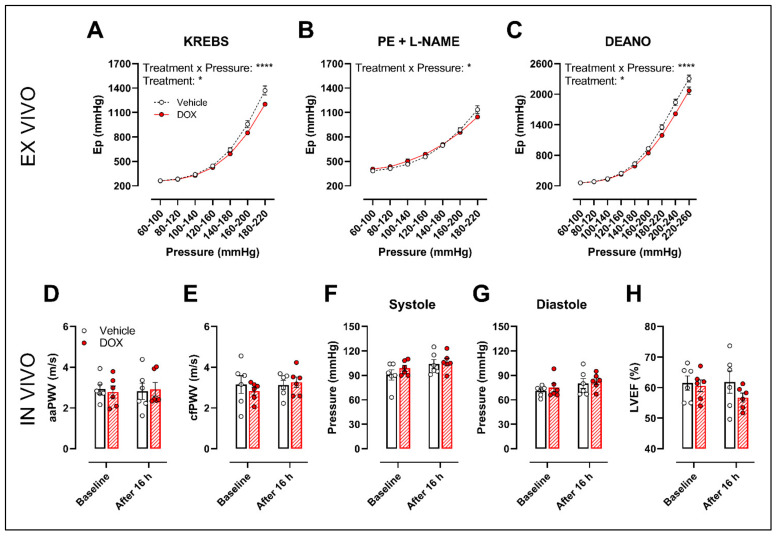
Evaluation of in and ex vivo aortic stiffness, 16 h after in vivo DOX administration. (**A**–**C**) Ex vivo evaluation of arterial stiffness with ROTSAC set-up. (**D**,**E**) In vivo evaluation of arterial stiffness with ultrasound imaging (**D**) and tonometry (**E**). (**F**) Systolic blood pressure measurements. (**G**) Diastolic blood pressure measurements. (**H**) Evaluation of LVEF with ultrasound imaging. Ep was lower in the DOX-treated group in a pressure-dependent way under Krebs Ringer conditions (**A**), but the effect was less pronounced in the presence of PE with L-NAME (**B**). Ep in response to DEANO was lower after DOX treatment (**C**). aaPWV (**D**), cfPWV (**E**), systolic blood pressure (**F**), diastolic blood pressure (**G**) and LVEF (**H**) were not affected after DOX treatment. For (**A–H**), repeated measures two-way ANOVA with Šidàk’s multiple comparisons test. *, **** *p* < 0.05, 0.0001; *n* = 6 for each group.

**Figure 7 ijms-22-12812-f007:**
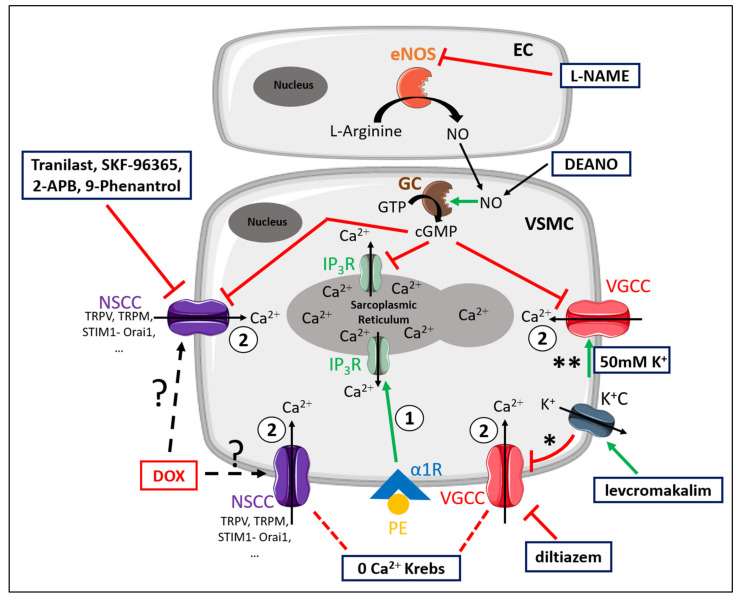
Schematic representation of the PE-elicited VSMC contraction response, including the compounds used to delineate the mechanisms of reduced VSMC contraction after DOX treatment. VSMC contraction through stimulation of α1-adrenergic receptors (α1R) with PE is characterised by two components—namely, a fast transient contraction (event 1) and a concomitant tonic contraction (event 2) [26]. Event 1: The fast transient contraction is determined by inositol trisphosphate (IP_3_)-mediated release of Ca^2+^ from the sarcoplasmic reticulum through IP_3_ receptors (IP_3_R) [26]. Event 2: The concomitant tonic contraction is determined by Ca^2+^ influx via VGCCs and NSCCs [26]. In both phases, cytoplasmic Ca^2+^ content increases, thereby forming a Ca^2+^–calmodulin complex [31], which activates the myosin light chain kinase, promotes the binding of actin to myosin and thus initiates contraction [31]. Production of NO in ECs inhibits Ca^2+^ release from the sarcoplasmic reticulum and VGCC- and NSCC-mediated Ca^2+^ influx in a cyclic guanosine monophosphate (cGMP)-dependent way, thereby continuously diminishing contraction [30]. This eNOS-mediated NO production can be inhibited by L-NAME. A 0 mM Ca^2+^ Krebs environment prevents Ca^2+^ influx, which results in exclusively IP_3_-mediated contraction. Diltiazem actively inhibits VGCCs, thereby inhibiting VGCC-mediated Ca^2+^ influx. Levcromakalim activates ATP-dependent K^+^ channels (K^+^C), resulting in K^+^ efflux, thereby causing a hyperpolarisation (event *) of the resting membrane potential and consequently deactivating VGCCs. In the presence of 50 mM K^+^, depolarisation will occur (event **), which will promote VGCC-mediated Ca^2+^ influx and subsequently leads to VSMC contraction. DEANO acts as an exogenous NO donor and inhibits VGCCs and NSCCs in a manner similar to NO. Members of the diverse family of NSCCs are TRPV, TRPM, STIM1-Orai1, channels, which can be inhibited with tranilast, SKF-96365, 2-APB and 9-phenantrol. DOX may interfere with these channels through an unclear mechanism. Red lines represent inhibition or deactivation, while green lines correspond to activation or stimulation.

## Data Availability

Not applicable.

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
