# Peer review of "Doxorubicin Impairs Smooth Muscle Cell Contraction: Novel Insights in Vascular Toxicity"

_ijms, 2021, doi:10.3390/ijms222312812_

Round 1
Reviewer 1 Report
This is a well-studied and well written manuscript. However, a lot of work has already been done and published previously on the effect of doxorubicin on VSMC function, contribution to arterial stiffness and cardiovascular outcome.
Author Response
Please find the answers to your raised questions and suggestions under "Reviewer 1" in the rebuttal document. The rebuttal document is provided as a Word file in the appendix.

Reviewer 2 Report
The author demonstrated the novel insight that DOX might interfere with VSMC functionality relying on non-selective cation channel-mediated Ca2+ influx, thereby contributing to arterial stiffness. Furthermore, an ex vivo model in aortic rings was successfully established to study the mechanism of VSMC function, which holds potential for future drug screening or mechanistic studies. With regards to the specific mechanism that the author suggested, we still have some concerns about the experimental design and then drawing conclusions. At present, this paper should not be taken into acceptance if the following issues could be addressed accordingly.
- As implied in the article, DOX may have a dose-dependent effect on VSMC functionality. The dosage used in this study was 4 mg/kg which was quite lower than the 12-15 mg/kg in clinical application. We suggest that more dosages and protocols, such as a long-term, low dosage, and continuous administration, should be taken into consideration and evaluated for clinical references.
- The results you conclude from figure 5 that the DOX may not have a notable inhibiting effect on VSMC’s VGCC activity are not persuasive enough, more supporting data and evidence are needed for further confirmation. If possible, pre-incubating DOX treated and vehicle groups with specific NSCC inhibitors to evaluate the effect of the NSCC dysfunction in aortic segments should be added to confirm the NSCC’s pivotal role in DOX treated VSMCs.
- Aside from the PE and K+, more cell contraction agonists and neurotransmitters should be tested to evaluate VSMCs’ responses in different mechanisms and environments.
Author Response
Please find the answers to your raised questions and suggestions under "Reviewer 2" in the rebuttal document. The rebuttal document is provided as a Word file in the appendix.

Reviewer 3 Report
The study by Bosman et al. investigated the role of doxorubicin (DOX) in vascular contraction. The authors used in vivo mice model for vascular contraction studies as well as an ex vivo model for pharmacological analysis with additional inhibitors of calcium channels. The present study reports decreased VSMC contraction under DOX treatment with coinciding endothelial dysfunction. The authors also hypothesise that DOX inhibits NSCC-mediated Ca2+-influx in aortic VSMCs. It is an interesting basic science study that contributes to our knowledge about DOX effects and mechanisms of action.
Comments
- In animals, like in humans, arterial stiffness could be evaluated in vivo. It is not clear, why the treated animals were not investigated in vivo in context of stiffness and blood pressure by using different doses of DOX. Please explain.
- The authors excluded several molecular mechanisms of action of DOX by using different inhibitors. Nevertheless, the main mechanism of action is not clearly explained. At the end of the discussion part, the authors hypothesise that DOX inhibits NSCC-mediated Ca2+-influx in aortic VSMCs. Are there any other possible mechanisms of action?
- Suggestion: It would be also relevant to add a hypothesised mechanism of DOX action on the schematic representation in Figure 7.
- Decreased VSMC contraction and de-stiffening found in your study may be followed by decreased blood pressure. Are their any data about blood pressure lowering effect by DOX?
- Is your study of clinical relevance?
Author Response
Please find the answers to your raised questions and suggestions under "Reviewer 3" in the rebuttal document. The rebuttal document is provided as a Word file in the appendix.

Round 2
Reviewer 1 Report
The manuscript can be accepted in the present form